# Supervisory Algorithm for Autonomous Hemodynamic Management Systems

**DOI:** 10.3390/s22020529

**Published:** 2022-01-11

**Authors:** Eric J. Snider, Saul J. Vega, Evan Ross, David Berard, Sofia I. Hernandez-Torres, Jose Salinas, Emily N. Boice

**Affiliations:** 1Engineering, Technology, and Automation Group, U.S. Army Institute of Surgical Research, JBSA Fort Sam Houston, San Antonio, TX 78234, USA; eric.j.snider3.civ@mail.mil (E.J.S.); saul.j.vega.ctr@mail.mil (S.J.V.); david.m.berard3.ctr@mail.mil (D.B.); sofia.i.hernandeztorres.ctr@mail.mil (S.I.H.-T.); jose.salinas4.civ@mail.mil (J.S.); 2Blood and Shock Resuscitation Group, United States Army Institute of Surgical Research, JBSA Fort Sam Houston, San Antonio, TX 78234, USA; evan.ross2.ctr@mail.mil

**Keywords:** hemorrhage, automation, closed-loop, algorithm, infusion, controller, adaptive

## Abstract

Future military conflicts will require new solutions to manage combat casualties. The use of automated medical systems can potentially address this need by streamlining and augmenting the delivery of medical care in both emergency and combat trauma environments. However, in many situations, these systems may need to operate in conjunction with other autonomous and semi-autonomous devices. Management of complex patients may require multiple automated systems operating simultaneously and potentially competing with each other. Supervisory controllers capable of harmonizing multiple closed-loop systems are thus essential before multiple automated medical systems can be deployed in managing complex medical situations. The objective for this study was to develop a Supervisory Algorithm for Casualty Management (SACM) that manages decisions and interplay between two automated systems designed for management of hemorrhage control and resuscitation: an automatic extremity tourniquet system and an adaptive resuscitation controller. SACM monitors the required physiological inputs for both systems and synchronizes each respective system as needed. We present a series of trauma experiments carried out in a physiologically relevant benchtop circulatory system in which SACM must recognize extremity or internal hemorrhage, activate the corresponding algorithm to apply a tourniquet, and then resuscitate back to the target pressure setpoint. SACM continues monitoring after the initial stabilization so that additional medical changes can be quickly identified and addressed, essential to extending automation algorithms past initial trauma resuscitation into extended monitoring. Overall, SACM is an important step in transitioning automated medical systems into emergency and combat trauma situations. Future work will address further interplay between these systems and integrate additional medical systems.

## 1. Introduction

Because they can automatically execute rapid adjustments in care in response to a patient’s needs, autonomous and semi-autonomous medical systems hold the promise of revolutionizing the practice and delivery of personalized medicine [1,2,3]. Rapid adjustments are particularly critical for patient management in the perioperative and intensive care setting, where closed-loop controllers have been developed for ventilator management [4,5,6] and fluid resuscitation [7,8,9,10], as well as pain control [11] and sedation [12]. In much the same way that a thermostat manages a household’s temperature by controlling an air conditioner or heater, these closed-loop medical systems capture a physiological variable (etCO_2_, SpO_2_, MAP, EEG, etc.), perform a calculation or other analysis on that data, then execute an action to adjust the variable towards a predefined setpoint.

Despite the sophistication of these autonomous and semi-autonomous systems, each algorithm normally addressed a single patient condition or intervention while remaining blind to the overall patient status. Indeed, as is often the case in the perioperative setting [13] and the intensive care unit [14,15], when a patient requires multiple interventions simultaneously the clinician will prioritize one line of treatment at the expense of another- in such a hypothetical situation in which two autonomous systems would be at odds, each algorithm lacks the ability to adjust its own performance to meet the needs of the other, potentially leading to catastrophic unintended consequences. Successful integration of multiple closed-loop systems, therefore, requires that the medical delivery team be capable of harmonizing the performance of the controllers to the benefit of the patient and adjudicating between the controllers when they conflict.

We argue that the next logical step in the field of autonomous medical systems is the development of supervisory algorithms that are specifically intended to integrate the performance of multiple closed-loop controllers towards a specific medical goal. To the best of our knowledge, supervisory algorithms for automating medical sub-systems have currently not been developed. As an initial proof of concept for a simple supervisory algorithm, we integrated the performance of our closed-loop tourniquet controller [16] and our closed-loop hemorrhage resuscitation controller [17] as deployed in a physiologically relevant benchtop model of hemorrhage and resuscitation.

The supervisory algorithm we describe here can recognize a simulated hemorrhage by sensing the loss of system pressure, activate a closed-loop tourniquet controller, and activate a closed-loop resuscitation algorithm to return the system to an arterial pressure setpoint. Importantly, the supervisory algorithm continues to monitor the patient for signs of tourniquet failure or ongoing hemorrhage and reacts accordingly. The supervisory algorithm is robust to many different patient presentations, as we detail here in five hemorrhage scenarios with increasing complexities.

## 2. Materials and Methods

### 2.1. Development of the Fluid Resuscitation Adaptive Controller

We previously developed a self-tightening, automated tourniquet [16]; but, in order to evaluate supervisory algorithm designs for multiple facets of hemodynamic management (see Section 2.2), we developed a closed-loop controller for fluid resuscitation. The adaptive resuscitation controller (ARC) received arterial pressure data as input, compared it to a desired mean arterial pressure (MAP) setpoint, and determined an appropriate infusion flow rate to efficiently stabilize arterial pressure to that setpoint (Figure 1A).

In general terms, for every cycle, ARC calculated an intermediate MAP setpoint between the currently measured MAP and the actual overall target MAP. Using a formula that varied with the infusate, the algorithm then calculated an infusion rate to achieve that intermediate MAP. ARC then activated an infusion pump at this flow rate, sampling MAP once per second, until the system reached the intermediate setpoint. At that point, a new intermediate MAP setpoint was calculated for the next cycle of the controller. This process of infusing to an intermediate setpoint and calculating a new one continued to loop until the measured MAP was within 99.5% of the overall target setpoint.

The ARC algorithm featured smarter flow rate decisions and adaptive capabilities for automated resuscitation by applying formulas derived from pressure-volume data sets of whole blood- and crystalloid-based resuscitation scenarios that we previously developed [17]. Using these data sets, polynomial curve fits were applied to derive normalized pressure–volume relationships that were utilized by ARC to determine the approximate fluid volume deficit at the current MAP and calculate a volume percent error from the setpoint. The volumetric infusion flow rate was determined by: $$ARC\;Flow\;Rate=\left(Q_{max}-Q_{min}\;\right)*\left(\%\;Volume\;Error\right)+Q_{min}$$
where the range of infusion flow rates ($Q_{max}-Q_{min}$) was multiplied by the volume deficit error and added to the minimum flow rate. This allowed the controller to apply quicker infusion rates when the pressure was further from the setpoint and to slow down as the target was approached.

In addition to calculating deviations from the MAP setpoint, ARC also featured an adaptive component that compared the time required to achieve a MAP increase for a given infusion rate (*t_actual_*) against the time predicted by the aforementioned experimental pressure-volume datasets. ARC used this discrepancy to calculate a correction factor for the infusion rate as: $$Correction\;Factor_{n+1}=Correction\;Factor_{n}*\;\frac{t_{actual}}{t_{predicted}}*\;Scaling\;Factor$$

The correction factor had an initial value of one and was adjusted during each ARC loop cycle, as per this formula. This adjustment allowed the ARC to increase or reduce the infusion rate as needed to better match the pressure-volume data sets. A scaling factor was added to the calculation with a default value of one, which could be modified higher or lower to increase or decrease the rate of change of the flow rate, respectively. In addition, the same discrepancy between actual and predicted times for changes in measured MAP was also used by ARC to detect potential hemorrhage while a resuscitation is ongoing: Anytime the actual time is twice as long as the predicted time, i.e., 100% error in time prediction, a “suspected hemorrhage” flag was raised, which automatically set the infusion to the maximum flow rate in an attempt to stabilize MAP in this scenario. ARC automatically resumed its regular monitoring operation soon after the criterion for “suspected hemorrhage” was no longer met.

The ARC algorithm is implemented in Python 3.8 and can execute on a standard personal computer. An arterial pressure input signal was collected by a pressure transducer (ABPDANT015PGAA5, Honeywell, Charlotte, NC, USA) and sampled at 50 Hz through a USB-connected data acquisition unit (LabJack Corporation, Lakewood, CO, USA). After calculating an infusion flow rate, the algorithm sent flow rate instructions to an infusion pump (Masterflex L/S, Masterflex Bioprocessing, Vernon Hills, IL, USA) through a serial port using open-source Python libraries (masterflex, Wyss Institute, Boston, MA, USA).

### 2.2. Development of the Supervisory Algorithm for Casualty Management

The Supervisory Algorithm for Casualty Management (SACM), in its current implementation, was designed to control and manage two independent algorithms: the self-tightening tourniquet (aTKT) and the adaptive resuscitation controller (ARC). It integrated the functions of both algorithms in such a manner as to allow them to semi-independently operate within their own areas of concern concurrently and without interfering with each other. Generally speaking, the aTKT sub-component could perform its function of arresting uncontrolled hemorrhage in an extremity [16], while ARC could perform fluid resuscitation should the subject drop their MAP value from hemorrhagic shock—the supervisory algorithm activated each as needed. The implementation of the supervisory algorithm proceeded in two stages: active intervention and continuous patient monitoring with additional intervention as needed.

The system, as designed, assumed that the subject had a bleeding extremity and was entering hemorrhagic shock (Figure 1B). Upon manually being started, SACM initiated the active intervention phase in which it sequentially acted to stop the bleed in the affected extremity and then moved to manage the hypovolemic shock. SACM first engaged the tourniquet to stop the extremity bleed by activating and yielding system control to the aTKT sub-component. The aTKT independently was able to tighten the tourniquet and verified that the hemorrhage had stopped. Once aTKT completed its operation, it returned control to SACM, which proceeded to activate the ARC sub-component. ARC automatically performed hemorrhagic shock resuscitation by infusing fluids until it detected that a target MAP had been reached. The ARC was allowed to complete its operation and then returned control to the supervisory algorithm. It was at this point that the system entered a “monitoring and intervention” phase.

During the monitoring stage, SACM monitored certain conditions and concurrently activated the aTKT and ARC in separate processes, as needed. To achieve this, the system monitored two logical conditions every 15 s: whether the tourniquet had become loose, thus allowing a re-bleed in the injured extremity, and whether the MAP had dropped to a clinically relevant low level (below 85 mmHg).

For detecting loosening of the tourniquet, SACM relied on input from aTKT, which was modified for this study to check for detection of air pressure oscillations in the cuff after tourniquet engagement (indicating a likely rebleed) as well as large drops in air pressure from the previous engagement (>33%, suggesting mechanical failure). If SACM detected that the tourniquet was loose, it intervened by initiating aTKT as a separate and independent process, so it ran concurrently with the supervisory algorithm. The aTKT was able to operate normally to retighten the tourniquet and stop the extremity bleed.

For assessing unexpected drops in MAP, SACM directly measured those values every 15 s using the same sensors as ARC. Just as with aTKT, should SACM detect a low MAP, it activated ARC as a separate and concurrent process, allowing it to continue its monitoring operation.

As all three algorithms (SACM, ARC, and aTKT) were designed to run concurrently during this monitoring phase, at any given time the system was capable of re-tightening a loose tourniquet while simultaneously performing a hemorrhagic shock fluid resuscitation. The SACM algorithm, just as aTKT and ARC, was also implemented in Python.

### 2.3. PhysioVessel Flow Loop Test Platform

A diagram of the flow loop system used for testing SACM is shown in Figure 2. A peristaltic pump (Masterflex L/S, Masterflex Bioprocessing, Vernon Hills, IL, USA) and water as the fluid supply were used to create pulsatile flow through the system to mimic physiological conditions. Pressure in the loop was monitored by a pressure transducer (PT1) (ICU Medical, San Clemente, CA, USA) positioned downstream from the pump. To enable complete flow occlusion by the tourniquet while permitting continuous flow driven by the circulatory pump, a bypass loop was integrated downstream from PT1, with a gate valve allowing tuning to facilitate various circulatory flow rates. A 112-mm-long, 110-mm-diameter phantom arm was molded from Ecoflex 00-10 silicone (Smooth-On Inc., Macungie, PA, USA), and included a 20-mm-diameter PVC pipe “bone” positioned centrally. A 4-mm-diameter hollow channel positioned 35 mm from the center was included in the mold to simulate an artery for the flow. Barbed fittings were secured in the “artery” at either end to connect the phantom arm in line with the loop tubing. A t-connector was used downstream from the phantom arm to connect a three-way valve that included a second PT (PT2, ICU Medical, San Clemente, CA, USA) and a bleed site. Upstream from the circulatory pump was a t-connector that integrated the PhysioVessel (PV) with the flow loop to provide both volume and pressure to the system. Hydrostatic pressure was supplied based on the height of the fluid column within the vessel. The relationship between the system volume and this height, and, thus, the pressure within the flow loop, can be precisely controlled to mimic normalized physiological relationships between volume and pressure seen in previous resuscitation studies [17]. Two peristaltic pumps (Masterflex L/S, Masterflex Bioprocessing, Vernon Hills, IL USA) were connected to the PV, one controlling resuscitation to the system and the other acting as an internal hemorrhage. Two different PV designs were used in testing initial ARC capabilities: (1) a whole blood PhysioVessel (PV_WB_) and (2) a crystalloid PhysioVessel (PV_C_). Each PV mimicked the normalized pressure-volume relationships of a porcine model for infusions of the corresponding fluids [17].

### 2.4. Adaptive Resuscitation Controller Experimental Design

Several experiments were run to evaluate the responsiveness of ARC under various conditions (Table 1), and three replicate infusions were conducted for each scenario. One set of test scenarios examined the effects of modifying the scaling factor used when calculating flow rate by ARC. Using PV_WB_, scaling factors of 0.5, 1 (baseline), and 2 were assessed with no hemorrhage occurring in the flow loop. A second set of testing scenarios observed how ARC responded while there was an ongoing hemorrhage occurring during resuscitation. The baseline scaling factor was used with the PV_WB_ and hemorrhage rates of 120, 240, and 360 mL/min were evaluated. Lastly, resuscitation with ARC using PV_C_ to alter the pressure-volume resuscitation curve was tested at a baseline scaling with no hemorrhage and a 240 mL/min hemorrhage rate.

### 2.5. Supervisory Algorithm Casualty Management (SACM) Scenario Testing

Integrating the functionality of the ARC and aTKT algorithms under SACM required us to test the algorithm’s robustness under a myriad of hemorrhage and resuscitation scenarios, for which the flow loop testing platform with the PV_WB_ was used. Five testing scenarios were conducted in triplicate (Table 2). Each scenario was broken into five time intervals or three for scenario #1. In all scenarios, first, the flow loop central MAP was destabilized with a simulated, manual hemorrhage that was stopped when MAP reached approximately 75 mmHg (Time Interval 0–1). Next, an extremity hemorrhage was introduced using a valve distal to the arm, at which point SACM was activated. The aTKT started tightening the tourniquet until the arm bleed was controlled (Time Interval 1–2). Then, the low central arterial pressure was re-stabilized by ARC up to approximately 95 mmHg (Time Interval 2–3).

While scenario #1 just evaluated these initial stabilization capabilities of SACM, the other scenarios evaluated how SACM handled additional challenges during the monitoring phase (Time Intervals 3–5). Two complications were evaluated. The first complication was loosening of the tourniquet, causing a re-introduction of an extremity hemorrhage for which SACM would need to re-engage aTKT to re-stabilize. The second complication was the introduction of a secondary internal hemorrhage site, which reduced central MAP but could not be stopped by tourniquet application, therefore requiring the ARC to restabilize central MAP. Each remaining scenario experienced two complications in all possible combinations (Table 1). For Scenario #5, internal hemorrhage (240 mL/min) continued while the tourniquet was loosened, resulting in a simultaneous occurrence of both complications, which was not observed in the other scenario options.

### 2.6. Data Analysis

A data acquisition system (PowerLab, ADInstruments, Sydney, Australia) recorded pressure waveform data using pressure transducers (ICU Medical, San Clemente, CA, USA) connected to a patient monitor (Infinity Delta XL Dräger, Lübeck, Germany) at two locations: in the central loop (PT1) and distal to the TKT application site (PT2). In addition, the data acquisition system recorded pneumatic tourniquet cuff pressure using an air pressure transducer (MPX5050DP, NXP Semiconductors, Eindhoven, The Netherlands). Infusion rates set by ARC during resuscitation were sampled at a rate of 1 Hz. Extremity bleed volume was collected in real time by accumulation in a vessel weighed using a force sensor (100 N, Mark-10, Copiague, NY, USA) and corresponding acquisition software (MesurGage Plus, Mark-10, Copiague, NY, USA). Internal hemorrhage volume was manually tracked by notes recorded during experiments as the flow rate was a constant when in use (240 mL/min).

For initial ARC evaluation studies, infusion flow rate heatmaps were generated in MATLAB (MathWorks, Natick, MA, USA) to compare the algorithm’s response to the test scenarios outlined in Table 1. Each set of three replicates were grouped and compared against each testing scenario. The *x*-axis represents time, and the shade of gray represents the flowrate recorded. All heatmaps were scaled according to the longest infusion duration for their respective PV (either whole blood or crystalloid derived) and the maximum infusion rate (1200 mL/min) observed.

For the various SACM scenarios tested, three replicates were performed for each with data recorded continuously. As data were collected through different means, time stamps were aligned for pressure and flow rate results, and each scenario time interval (see Semi-Autonomous Casualty Management Testing Scenarios Methods Section) was divided and the time range for each replicate bin was set from n − 1 to n to confine the data into three or five bins (three bins for scenario #1, five bins for the rest). Next, fluid balance was calculated as the total infused volume minus the total hemorrhage to evaluate how close the system returned to its initial value. Replicates were plotted using MatPlotLib (v3.4.3 [18], Python v3.9) for each of the five parameter types (MAP Distal Arm, TKT Pressure, Central MAP, Infusion Rate, and Fluid Balance) for evaluation of SACM performance in each testing situation.

## 3. Results

### 3.1. Experimental Evaluation of the Adaptive Resuscitative Controller (ARC)

We first evaluated the capabilities and performance of the ARC infusion system for stabilizing hypovolemic situations simulated in the PV flow loop platform. In the initial ARC setup using the baseline correction factor calculation for infusion rate, the system performed as expected when using PV_WB_, with initially high flow rates reduced to near minimum value as the target setpoint was reached (Figure 3A). Similar results were evident with PV_C_ setups (Figure 3B), except with flow rates not dropping as low and requiring more time to reach the setpoint, as the fluid capacity of PV_C_ was more than 2× the whole blood setup [17]. We had similar performance for the system when we changed the scaling factor to 0.5 (Figure 3C); however, when the scaling factor was increased to 2, the flow rate increased and held at the maximum 1200 mL/min flow rate almost immediately (Figure 3D). This highlights the tunability of the system that can be modulated to meet the end user’s desired resuscitation rate.

Oftentimes in trauma scenarios, especially in combat casualty care and austere environments, addressing a sudden hemorrhage immediately may not be possible, so it may be critical to temporarily continue stabilizing the casualty during an ongoing hemorrhage. We simulated such a scenario by utilizing hemorrhage rates of three different severities during resuscitation: 120, 240, and 360 mL/min. At a 240-mL/min hemorrhage rate, the ARC infusion responsiveness was more variable between replicates (Figure 4A). Much of this variability was switching in and out of hemorrhage detection (100% error in time predictions, see methods), which was less pronounced for the PV_WB_ compared to the PV_C_ results (Figure 4B). High flow rates remained when using PV_C_, but not all replicates reached the hemorrhage detection threshold. For reduced 120-mL/min hemorrhage rates, flow rates and ARC trends were more consistent with non-hemorrhage trends, indicating the 100% error threshold was rarely reached at these slower rates (Figure 4C). Conversely, the opposite trend was evident at higher 360-mL/min flow rates where hemorrhage detection remained for nearly the entire ARC infusion (Figure 4D).

### 3.2. Overview of SACM Algorithm and Its Utility for Initial Hemodynamic Stabilization

As a first example of the utility of the SACM within the PV_WB_ flow loop platform, we evaluated its ability to initially stabilize a trauma patient with minimal complications (SACM Testing Scenario #1, Table 2). In this scenario (Figure 5), the flow loop platform acted as the “patient”, presenting with an initial hemorrhage that caused the MAP to fall to 75 mmHg (Figure 5A, Interval 0–1). Then, an extremity bleed was induced (bleed site from the silicone arm analogue). At this point, SACM was initiated, which initially activated aTKT (Figure 5A, Interval 1–2). The activation of aTKT applied incremental increasing tourniquet pressures (Figure 5B, Interval 1–2), which caused the MAP distal to the arm to fall to near zero; however, the central MAP in the system remained near 75 mmHg (Figure 5C, Interval 1–2). Next, SACM initiated ARC to re-stabilize the system at 95 mmHg by activating the infusion pump to add fluid to the PV_WB_ (Figure 5D, Interval 2–3). The fluid balance across the scenario showed a loss of fluid during the initial hemorrhage and extremity bleed (Figure 5E, Interval 0–1) and a recovery during resuscitation (Interval 2–3). The tourniquet pressure remained elevated post-aTKT application as aTKT was fully engaged and minimally loosened.

### 3.3. SACM Utility for Extended Patient Management

As a proof of concept for extended patient monitoring and management, SACM was tested in additional scenarios that introduced various complications (Table 2). SACM Testing Scenario #2 (n = 3) followed Scenario #1 initially but included complications of the tourniquet loosening (Figure 6). This scenario was consistent with environmental situations that result in loosening of the device, such as patient motion during medical evacuation.

As with Scenario #1, the initial hemorrhage reduced the MAP distal to the arm and the central MAP to around 75 mmHg (Figure 6A, Interval 0–1). The extremity bleed was initiated and SACM activated aTKT to engage the tourniquet (Figure 6B, Inteval 1–2), which reduced the MAP distal to the arm to near zero, while the central MAP was stabilized (Figure 6A,C, Interval 1–2). The different replicates shown in the tourniquet pressure panel reached different final cuff pressures, but all reached a plateau that signaled the occlusion of fluid flow. Subsequently, SACM signaled ARC for resuscitation (Figure 6D, Interval 2–3), adding fluid to the PV_WB_ (Figure 6E, Interval 2–3), which restored central MAP to near baseline levels (Figure 6C, Interval 2–3). However, in Scenario #2 the tourniquet was manually loosened, allowing re-perfusion into the arm (Figure 6A, Interval 3–4). This triggered the monitoring portion of SACM to re-engage aTKT to tighten the tourniquet (Figure 6B, Interval 3–4), reducing the MAP distal to the arm. Upon re-stabilization, the tourniquet was once again manually loosened (Figure 6B, Interval 4–5), fluid flow was detected at the bleed site past the arm (Figure 6A, Interval 4–5), and aTKT was re-engaged. After this second re-engagement of aTKT, the central MAP recovered to near baseline levels (Figure 6C, Interval 4–5). Minor fluctuations in the fluid balance showed small volumes of fluid loss during tourniquet loosening (Figure 6E, Interval 3–5) but the ARC monitoring did not detect a sufficiently large drop in central MAP to trigger another round of resuscitation.

Scenario #3 opened with the same process of events (n = 3): initial hemorrhage, extremity bleed, aTKT activation and tourniquet tightening, and ARC-driven resuscitation to bring the central MAP back up to a baseline of 95 mmHg (Figure 7, Interval 0–2). However, in this scenario, an internal hemorrhage was added as a complication. In trauma settings, patients can present with consistently dropping MAP values, which cannot be stopped by tourniquet, indicating the possibility of an internal hemorrhage. The monitoring portion of SACM tracked central MAP values and, during this simulated internal hemorrhage, it detected a falling arterial pressure (Figure 7C, Interval 3–4), resulting in the activation of ARC for fluid resuscitation. With the addition of fluid to the PV_WB_ by ARC (Figure 7D, Interval 3–4), the fluid balance was brought back to near baseline (Figure 7E, Interval 3–4), and central MAP was brought back up to 95 mmHg. As in a real-world scenario, where the internal bleed was not immediately found and hemorrhaging continued, the central MAP fell for a third time (Figure 7C, Interval 4–5), which was detected by SACM once again, reactivating ARC. Fluid was added to stabilize for a third time, and the internal bleed was stopped, allowing the loop to settle at 95 mmHg.

To further test the capabilities of the monitoring portion of SACM, a scenario was devised to follow the initial set of events and added both a loosened tourniquet and an internal hemorrhage (Figure 8). For SACM Testing Scenario #4, the initial hemorrhage, extremity bleed, aTKT intervention, and initial resuscitation (Figure 8A, Interval 0–3) all proceeded identically to other scenarios (n = 3). In a single replicate, the tourniquet cuff pressure reached a higher plateau (Figure 8B, Interval 2–3), but all three runs occluded flow at the bleed site and allowed the central MAP to return to baseline after execution of ARC (Figure 8C, Interval 2–3). Next, the manual loosening of the tourniquet was detected by SACM (Figure 8A, Interval 3–4), which initiated aTKT retightening protocol (Figure 8B, Interval 3–4). This aTKT re-engagement stabilized central MAP (Figure 8C, Interval 3–4). An internal hemorrhage was then added to further test the monitoring function by SACM, which successfully detected the reduction in the central MAP value (Figure 8C, Interval 4–5) and re-activated ARC. This resulted in fluid being added to the PV_WB_ (Figure 8D, Interval 4–5), pressure being returned to the setpoint (Figure 8C, Interval 4–5), and an overall recovery of the loss of fluid (Figure 8E, Interval 4–5).

For the final evaluation of SACM, Scenario #5 tested (n = 3) whether SACM could detect a loosened tourniquet while an active internal bleed was underway. The initial hemorrhage, extremity bleed, aTKT engagement, and initial resuscitation all proceeded normally (Figure 9, Interval 0–3). Upon stabilization of central MAP to 95 mmHg, an internal hemorrhage was induced (Figure 9A, Interval 3–4), and the ARC protocol was initiated to restabilize MAP at 95 mmHg (Figure 9D, Interval 3–4). Upon ARC completion, fluid balance was restored for a short period of time (Figure 9E, Interval 3–4) followed by loosening of the tourniquet. SACM, while the internal hemorrhage was still active, detected loosening of the tourniquet and re-engaged aTKT tightening protocol (Figure 9B, Interval 4–5). As central MAP was below the 85 mmHg threshold due to the ongoing internal hemorrhage (Figure 9C, Interval 4–5), SACM activated a third ARC resuscitation (Figure 9D, Interval 4–5). Central MAP in the loop recovered to 95 mmHg and fluid balance was restored to slightly higher than initial levels (Figure 9E, Interval 4–5).

## 4. Discussion

Medical automation holds the potential of simplifying the delivery of both civilian and military emergency medical care by making rapid patient management decisions, thus cognitively offloading caregivers in situations where they can be overwhelmed. Ventilator, resuscitation, and anesthesia medical systems are just a few systems with recent advancements toward closed-loop, fully autonomous operation. However, as multiple automated systems emerge, control systems and algorithms will be needed to coordinate operation across these systems or applications to ensure physiological data are being tracked and managed appropriately as the patient status changes. Here, we highlighted the utility of SACM, a control algorithm that monitors relevant inputs and controls decisions being made across two closed-loop medical systems, aTKT and ARC.

We first highlighted ARC, which controls hemorrhagic resuscitation with minimal user input. ARC compares and fine-tunes flow rates based on approximate volume deficit and differences relative to trained pressure-volume relationships. The correction factor sensitivity can be adjusted to increase or decrease the intensity of flow rate changes to reach the setpoint. The precise speed and approach taken during resuscitation may vary based on infusate and patient condition, which can be easily accounted for with ARC. In addition, we incorporated an active hemorrhage detection feature within ARC that activates if it takes twice as long to accomplish a pressure increase at a given infusion flow rate than trained data comparisons predicted. The baseline correction factor sensitivity resulted in the slowest overall resuscitation for both PV_WB_ and PV_C_. Unexpectedly, a 0.5× correction factor sensitivity in PV_WB_ accomplished a faster resuscitation than baseline. This is, however, reasonable since the correction factor sensitivity modifies the *magnitude* of changes to the flow rate, whether those changes are positive or negative. As pressure nears the setpoint and ARC calls for the flow rate to be reduced, a 0.5× sensitivity decreases the magnitude of each reduction resulting in a slightly higher flow rate than baseline over time. Alternatively, a sensitivity of 2× caused a significantly large initial increase in the flow rate that was maintained until the setpoint was reached. At a baseline sensitivity, the slower hemorrhage rate of 120 mL/min was difficult to detect in 2/3 repetitions until the end when infusion flow rates were substantially reduced, with larger hemorrhages proving to be more easily detected. As with the infusion rate, the ARC hemorrhage detection threshold can be easily adjusted to increase or decrease sensitivity depending on the scenario. These results demonstrate the degree of tuning available with ARC and its capability to be adapted to the end user’s most preferred resuscitation profile. In resource-austere conditions, such as mass casualty events, delayed evacuation in wilderness medicine, and prolonged field care scenarios on the future battlefield, access to definitive surgical care of active bleeding will be unavailable for long periods of time. Under these conditions, a resuscitative system must be able to manage patients who repeatedly become hemodynamically unstable despite ongoing fluid infusion, as this is an indication that the hemorrhage remains uncontrolled.

Next, we merged the aTKT and ARC autonomous systems under one master controller to highlight the interplay and cross-talk required between medical systems such as these as automation becomes more prevalent in emergency medicine. The master controller, termed SACM, acts as a decision-making controller as to when each sub-system should be utilized. If MAP falls below a threshold, this signals that ARC is needed to re-stabilize at a healthy MAP level. If oscillations are detected in the tourniquet cuff, the tourniquet has loosened and needs to be re-engaged. Both of these are constantly monitored to ensure conditions adjustable by each sub-unit are stable.

The utility of a system like SACM overseeing sensors and sub-unit decisions was demonstrated through several scenarios where the closed-loop systems initially hemodynamically stabilized the testing system and continued to monitor and make decisions as complications occurred. For instance, tourniquets loosening or slipping resulting in an increased hemorrhage was tracked by SACM and within 30 s had identified a re-bleed from the arm site and began re-tightening the tourniquet. Initial design focused on re-tightening if the tourniquet loosened, but SACM could additionally be adapted to periodically loosen, providing re-infusion to the extremity to help manage limb survival [19,20,21]. Next, we highlighted the effect a secondary internal hemorrhage had on SACM. This resulted in a reduction in blood pressure that the aTKT was not capable of stopping, leading to resuscitation and indication of a secondary hemorrhage to the end user by ARC. Lastly, combination injuries were tested on SACM. Tourniquet loosening was immediately detected by SACM and aTKT re-applied the tourniquet to stop additional extremity hemorrhage. As blood pressure continued to decrease, the internal hemorrhage was detected and MAP restabilized via ARC.

Quick control and monitoring of multiple interconnected sub-units as contained in SACM are critical for optimal utility of these systems, especially as more advanced artificial intelligence is developed. For instance, closed-loop systems are in development for ventilators [4,5,6] and anesthesia [12] devices as well as many other technologies. In trauma care, there is a precise interplay between these systems similar to how SACM monitors hemodynamic management that will need to be continuously monitored [13,14]. The complexity and acuity of patients in critical care and perioperative settings often require that the medical care team prioritize one aspect of patient management over another: As an example, resuscitation goals and ventilator management goals may conflict, requiring an adjustment in the fluid management strategy to safely maintain the patient’s respiratory status. In a setting where multiple systems are present but disconnected from one another, a stand-alone, closed-loop resuscitation system and a stand-alone, closed-loop ventilator system would work against each other, with each system negatively impacting the function of the other with potentially deadly results. Supervisory algorithms could solve this problem by adjudicating between two or more conflicting systems with the goal of harmonizing their function towards patient safety. The challenge of programming this sort of supervisory algorithm is not trivial, given the vast array of potential circumstances where stand-alone systems may conflict; because the rules governing a hypothetical supervisory algorithm’s decisions for complex patient situations are unclear, additional research is needed to identify clinical guidelines and best practices to provide a decision-framework for how hypothetical supervisory algorithms would go about managing multiple closed-loop systems.

One shortcoming towards proper development of these systems and supervisory controllers such as SACM is advanced physiological monitoring. More sensitive, early indicators of complications are needed, such as compensatory reserve index (CRI) [22,23,24]. These advanced, early indicators will be essential to ensuring decisions are being made appropriately and early enough to keep the patient stable. Further, non-invasive sensing is critical for initial resuscitation efforts as arterial line placement is challenging in austere, military medicine situations. Research is being conducted by many groups to alleviate this and could be paired with advance detection metrics to enhance development of SACM going forward [25,26,27].

In addition to integration of advanced physiological metrics and non-invasive sensing, next steps for this work will further develop the decision matrix required for SACM in two ways. First, a more robust supervisory algorithm can be developed using other architectures, such as an inference engine. This will allow for designing a supervisory algorithm capable of managing a larger number of sub-components along with more complex and possibly conflicting logical decisions for clinical interventions than the very limited set of systems that the current implementation of SACM can handle. Such a design will likely be essential as SACM encompasses additional closed-loop systems and handles more complex trauma scenarios. Second, next steps will translate SACM and closed-loop system testing into large animal model testing. The PV flow-loop platform is optimal for initial troubleshooting but incorporating additional systems and biological complexities will only be possible with animal testing and will inform subsequent SACM improvements.

In conclusion, automating various medical systems can and will simplify emergency and military medicine, but overarching algorithms such as SACM demonstrated in this study will be essential to processing physiological inputs for when and how multiple automated systems should act.

## Figures and Tables

**Figure 1 sensors-22-00529-f001:**
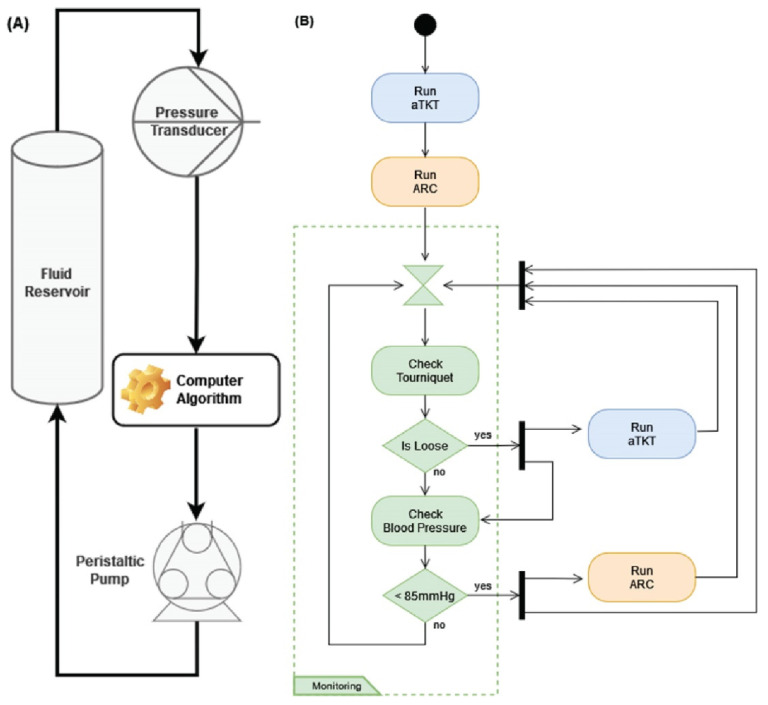
Description of the Closed-Loop Algorithm Architecture. (**A**) ARC for fluid resuscitation. A pressure input is sent to the controller and compared with the desired setpoint; the flow rate is adjusted based on distance from the setpoint. The controller also adapts by comparing performance to trained data sets and modifying the flow rate change amount accordingly. (**B**) SACM combines a previously developed auto-tourniquet with ARC. Decisions and interplay between the two sub-units are controlled by SACM, as shown.

**Figure 2 sensors-22-00529-f002:**
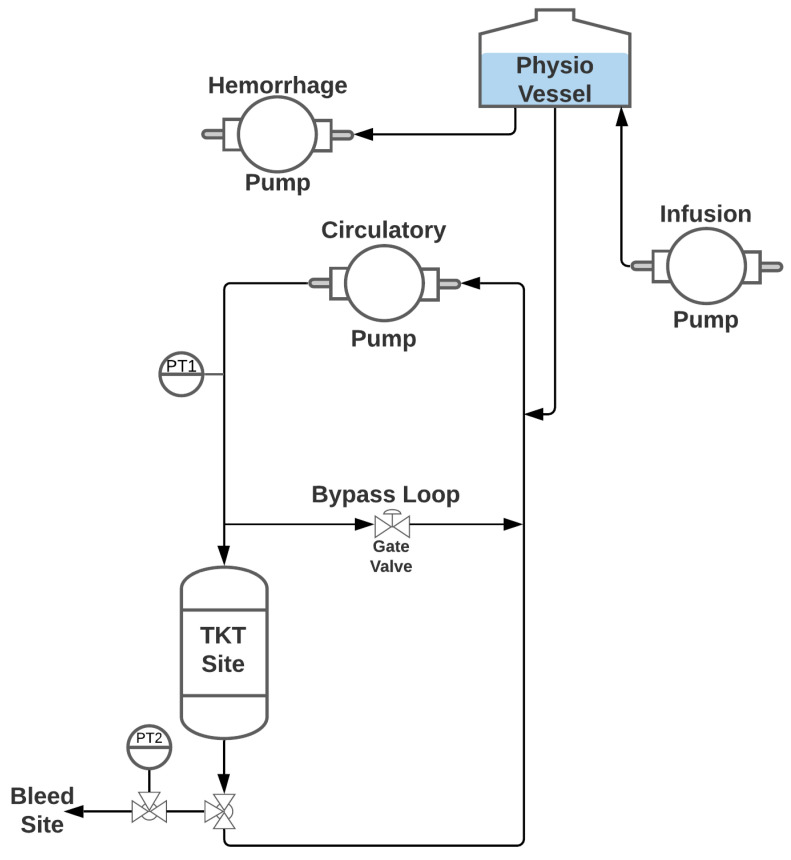
Diagram of the PhysioVessel flow loop test platform. A peristaltic pump (labeled Circulatory Pump) provided pulsatile flow, which was read by a pressure transducer (PT1) mimicking arterial line placement. Flow passed through an arm analogue made of silicone where a tourniquet was placed, and an extremity bleed was started distal to the arm; a second pressure readout was collected at the hemorrhage site (PT2). Arm and bypass flow streams merged upstream from the PhysioVessel, followed by completing the loop at the circulatory pump. Internal hemorrhage was pulled directly from the PhysioVessel by a peristaltic pump (labeled Hemorrhage Pump), and resuscitation was managed by a third peristaltic pump (labeled Infusion Pump) by infusing directly into the PhysioVessel.

**Figure 3 sensors-22-00529-f003:**
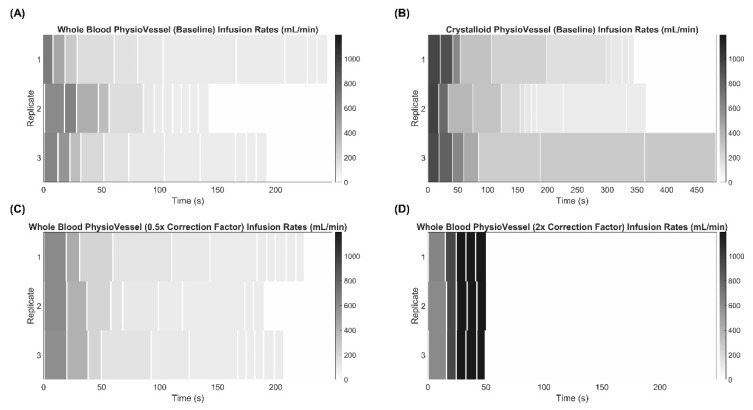
Effect of Adaptivity on the Adaptive Resuscitation Controller (ARC). Flow rate determined by ARC vs. time starting after initial hemorrhage to approximately 75 mmHg until the target of 95 mmHg was reached. Three technical replicates are shown as separate rows to illustrate how step sizes and flow rate changes varied. Heat map intensities range from 1200 to 0 mL/min for each, and time axes are equal for corresponding PV models to highlight the difference in rates to stabilize. Adaptive resuscitation controller results are shown first for the scaling factor set at 1 for (**A**) PV_WB_ and (**B**) PV_C_. In addition, results are shown for the scaling factor (**C**) reduced to 0.5 or (**D**) increased to 2 using PV_WB_ to highlight the adaptability of the system.

**Figure 4 sensors-22-00529-f004:**
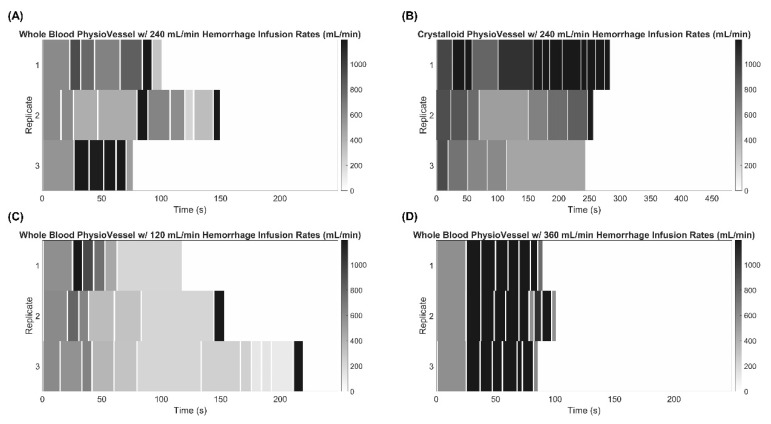
Effect of internal hemorrhage on the Adaptive Resuscitation Controller (ARC). Flow rate determined by ARC vs. time starting after initial hemorrhage to approximately 75 mmHg until the target of 95 mmHg was reached with a continuous internal hemorrhage as indicated. Results are shown for an internal hemorrhage rate of 240 mL/min during resuscitation using (**A**) PV_WB_ and (**B**) PV_C_, (**C**) 120 mL/min hemorrhage rate using PV_WB_, or (**D**) 360 mL/min hemorrhage rate using PV_WB_. Three technical replicates are shown as separate rows to illustrate how step sizes and flow rate changes varied. Heat map intensities range from 1200 to 0 mL/min for each, and time axes are equal for corresponding PV models to highlight the difference in rates to stabilize.

**Figure 5 sensors-22-00529-f005:**
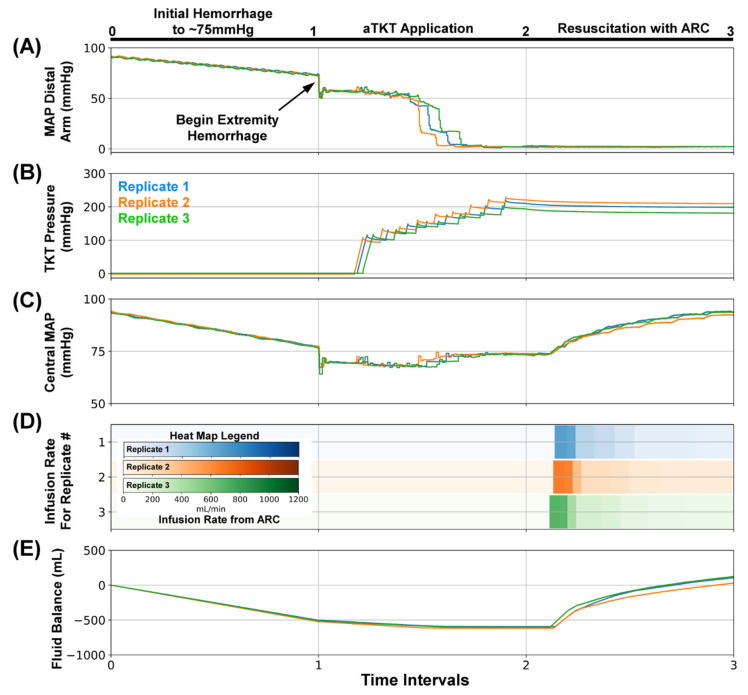
SACM Testing Scenario #1. After (0–1), the initial hemorrhage reduced blood pressure to approximately 75 mmHg, (1–2) extremity hemorrhage was stopped by aTKT, (2–3) followed by ARC-based resuscitation to 95 mmHg. Aligned results are shown for (**A**) MAP distal to the bleed site (mmHg, PT2), (**B**)Tourniquet Cuff pressure (mmHg), (**C**) Central MAP (mmHg, PT1), (**D**) Resuscitation flow rate (mL/min), and (**E**) overall system fluid balance (Infused Volume—Hemorrhage Volume, mL). Three individual replicates are shown to illustrate the similarities and difference between each replicate run.

**Figure 6 sensors-22-00529-f006:**
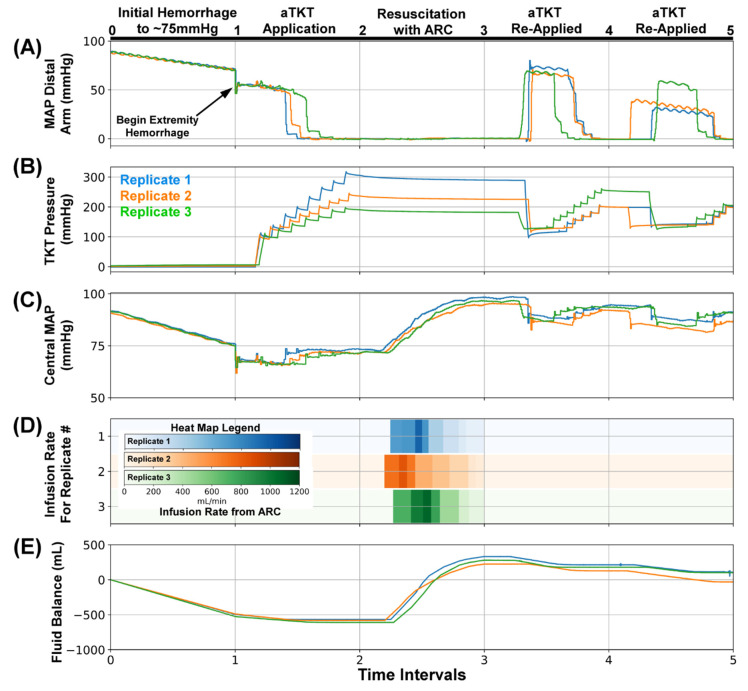
SACM Testing Scenario #2. After (0–1), the initial hemorrhage reduced blood pressure to approximately 75 mmHg, (1–2) extremity hemorrhage was stopped by the aTKT, (2–3) and resuscitation restored pressure to 95 mmHg by ARC. Next, (3–4) tourniquet cuff pressure was slightly reduced to mimic loosening, which was detected by SACM and re-inflated by aTKT; (4–5) a similar tourniquet loosening was repeated afterwards. Aligned results are shown for (**A**) MAP distal to the bleed site (mmHg, PT2), (**B**) Tourniquet Cuff pressure (mmHg), (**C**) Central MAP (mmHg, PT1), (**D**) Resuscitation flow rate (mL/min), and (**E**) overall system fluid balance (Infused Volume—Hemorrhage Volume, mL). Three individual replicates are shown to illustrate the similarities and difference between each replicate run.

**Figure 7 sensors-22-00529-f007:**
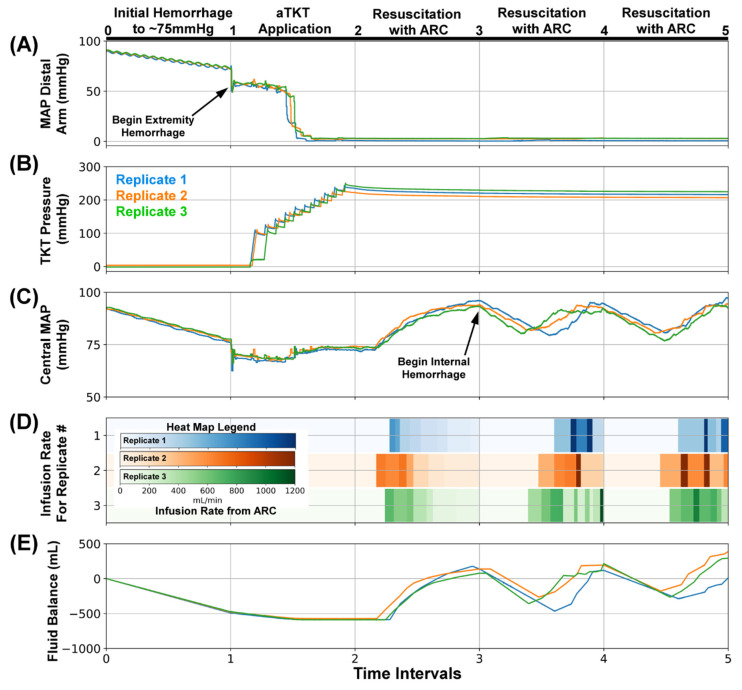
SACM Testing Scenario #3. After (0–1), the initial hemorrhage reduced blood pressure to approximately 75 mmHg, (1–2) extremity hemorrhage was stopped by aTKT, (2–3) and resuscitation restored pressure to 95 mmHg by ARC. Next, (3–4) an internal hemorrhage (240 mL/min) was induced, causing central MAP to fall to 85 mmHg in which SACM detected the reduction and resuscitated after checking the tourniquet cuff pressure did not slip; (4–5) a similar internal hemorrhage (240 mL/min) test scenario was repeated. Aligned results are shown for (**A**) MAP distal to the bleed site (mmHg, PT2), (**B**) Tourniquet Cuff pressure (mmHg), (**C**) Central MAP (mmHg, PT1), (**D**) Resuscitation flow rate (mL/min), and (**E**) overall system fluid balance (Infused Volume—Hemorrhage Volume, mL). Three individual replicates are shown to illustrate the similarities and difference between each replicate run.

**Figure 8 sensors-22-00529-f008:**
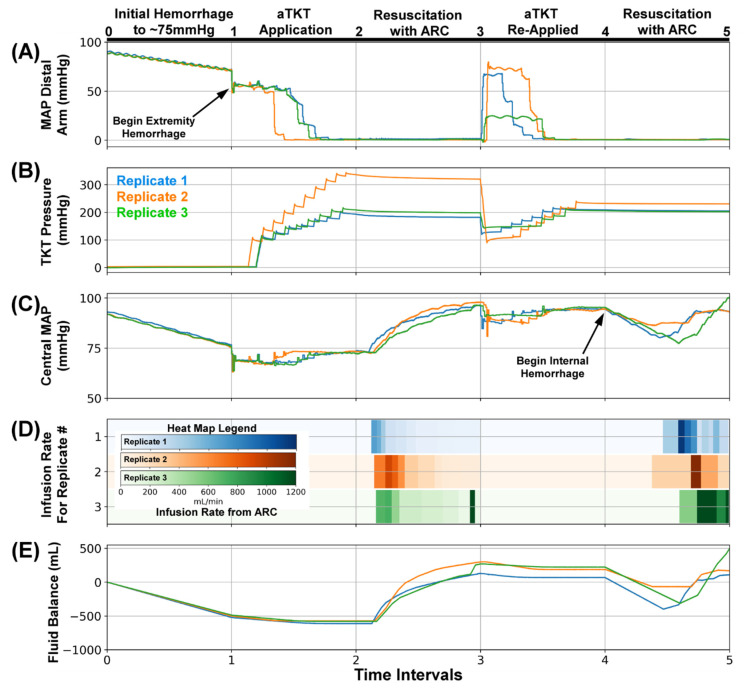
SACM Testing Scenario #4. After (0–1), the initial hemorrhage reduced blood pressure to approximately 75 mmHg, (1–2) extremity hemorrhage was stopped by aTKT, (2–3) resuscitation restored pressure to 95 mmHg by ARC. Next, (3–4) and tourniquet cuff pressure was slightly reduced to mimic loosening, which was detected by SACM and re-inflated by aTKT. Lastly, (4–5) an internal hemorrhage (240 mL/min) was induced, causing central MAP to fall to 85 mmHg in which SACM detected the reduction and resuscitated after checking the tourniquet cuff pressure did not slip. Aligned results are shown for (**A**) MAP distal to the bleed site (mmHg, PT2), (**B**) Tourniquet Cuff pressure (mmHg), (**C**) Central MAP (mmHg, PT1), (**D**) Resuscitation flow rate (mL/min), and (**E**) overall system fluid balance (Infused Volume—Hemorrhage Volume, mL). Three individual replicates are shown to illustrate the similarities and difference between each replicate run.

**Figure 9 sensors-22-00529-f009:**
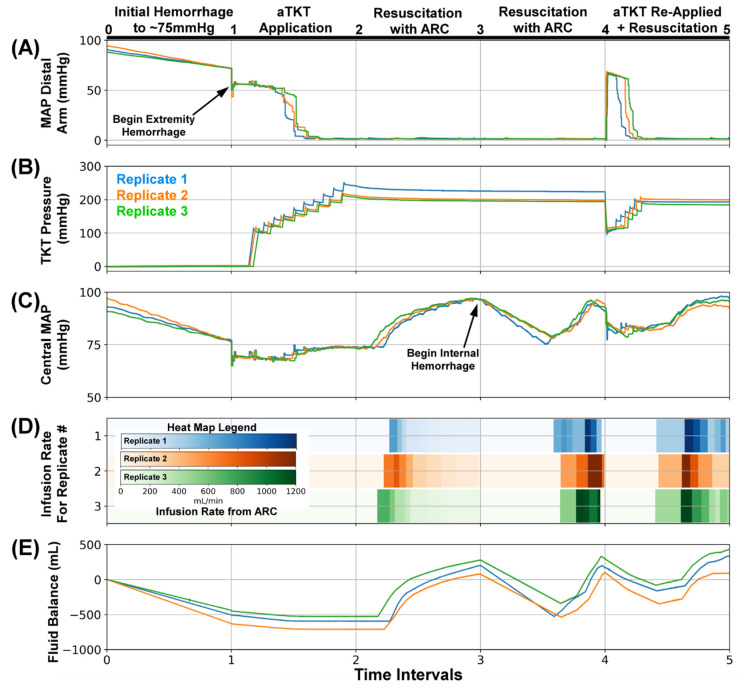
SACM Testing Scenario #5. After (0–1), the initial hemorrhage reduced blood pressure to approximately 75 mmHg, (1–2) extremity hemorrhage was stopped by aTKT, (2–3) and resuscitation restored pressure to 95 mmHg by ARC. Next, (3–4) an internal hemorrhage (240 mL/min) was induced, causing central MAP to fall to 85 mmHg in which SACM detected the reduction and resuscitated after checking the tourniquet cuff pressure did not slip. Lastly, (4–5) while internal hemorrhage (240 mL/min) continued, tourniquet cuff pressure was slightly reduced to mimic loosening, which was detected by SACM and re-inflated by aTKT algorithm, followed by resuscitation up to 95 mmHg. Aligned results are shown for (**A**) MAP distal to the bleed site (mmHg, PT2), (**B**) Tourniquet Cuff pressure (mmHg), (**C**) Central MAP (mmHg, PT1), (**D**) Resuscitation flow rate (mL/min), and (**E**) overall system fluid balance (Infused Volume—Hemorrhage Volume, mL). Three individual replicates are shown to illustrate the similarities and difference between each replicate run.

**Table 1 sensors-22-00529-t001:** Experimental Design for Evaluating the Adaptive Resuscitation Controller. Testing conditions for demonstrating the ARC capabilities using various hemorrhage rates and scaling factors, as well as evaluation with the whole blood and crystalloid PhysioVessel. Triplicate tests were performed at each scenario.

Adaptive Resuscitation Controller (ARC) Testing Scenarios		
**Whole Blood** **PhysioVessel (PV_WB_)**	Scaling factor adjustment	0.5× Scaling Factor
		Scaling Factor = 1 (Baseline)
		2× Scaling Factor
	Baseline scaling factor with internal hemorrhage	120 mL/min
		240 mL/min
		360 mL/min
**Crystalloid** **PhysioVessel (PV_C_)**	Baseline scaling factor	
	Internal Hemorrhage at 240 mL/min	

**Table 2 sensors-22-00529-t002:** Experimental Design for Evaluating SACM. Scenarios or case studies for demonstrating the utility of having SACM controlling multiple autonomous system sub-units. Five scenarios in total were evaluated with different internal or extremity hemorrhages, as shown. Triplicate tests were performed at each scenario.

Testing Scenario	Normalized Time Intervals for Each Scenario				
	(0–1)	(1–2)	(2–3)	(3–4)	(4–5)
**#1**	Initial Hemorrhage	Extremity Bleed + aTKT	ARC Resuscitation		
**#2**	Initial Hemorrhage	Extremity Bleed + aTKT	ARC Resuscitation	Internal Hemorrhage + ARC Resuscitation	Internal Hemorrhage + ARC Resuscitation
**#3**	Initial Hemorrhage	Extremity Bleed + aTKT	ARC Resuscitation	Loosen Tourniquet + aTKT	Loosen Tourniquet + aTKT
**#4**	Initial Hemorrhage	Extremity Bleed + aTKT	ARC Resuscitation	Loosen Tourniquet + aTKT	Internal Hemorrhage + ARC Resuscitation
**#5**	Initial Hemorrhage	Extremity Bleed + aTKT	ARC Resuscitation	Internal Hemorrhage + ARC Resuscitation	Loosen Tourniquet + aTKT

## Data Availability

The datasets generated during and/or analyzed during the current study are available from the corresponding author upon reasonable request.

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
