# Peer review of "Supervisory Algorithm for Autonomous Hemodynamic Management Systems"

_sensors, 2022, doi:10.3390/s22020529_

Round 1
Reviewer 1 Report
The use of automated medical systems can potentially address this need by streamlining and augmenting the delivery of medical care in both emergency and combat trauma environments. However, in many situations these systems may need to operate in conjunction with other autonomous and semiautonomous devices , this paper introduce a Supervisory Algorithm for Casualty Management (SACM). However, I thought this work has some deficiencies. The paper needs further improvement in terms of quality. Detailed comments are listed below:
--Section1: In terms of algorithms studied by previous researchers, the author did not introduce this. Is there other scholars done similar research on supervisory algorithm?.
--Section1: In the literature review part, the author introduced the current research status of intelligent unmanned mining, the literature review should be concise, try to draw general conclusions through abstracting and summarizing.
--Section1: Why did the author introduce the research algorithm of this article? In comparison, what are the advantages of this algorithm? Can it solve the problems of other methods mentioned by the author in this part?
--Section2.1: Please consider adding a description of SACM and analysis principles and processes in this section
--Section2.1: The author mentioned “The ARC algorithm featured smarter flow rate decisions and adaptive capabilities for automated resuscitation by applying formulas derived from pressure-volume data sets of whole blood- and crystalloid-based resuscitation scenarios that we have previously developed ”.I don’t think that readers will read the literature describing the data. Here, the author is asked to add the relevance of the data and demonstrate the rationality of the data.
--Section2.1: How is the correction factor determined? Please make it clearly.
--Section2.3: In the PhysioVessel flow loop test platform, for the blood circulation test process, what is the internal fluid? If other experimental liquids are considered, will it affect the experimental results?
--Section2: How is the internal flow rate controlled in the experimental device designed by the author? Can the author add corresponding diagrams of human blood circulation?
--Section2.5: In the Supervisory Algorithm Casualty Management (SACM) Scenario Testing process, how to set the parameters of different scenarios
--The description of Figure 3 and Figure 4 is not clear, please introduce the elements in the figure in detail. In addition, the image quality is relatively fuzzy, and the font size is also problematic. Please modify it and improve it.
--Section3: After comparing the ability and performance of the infusion system in the PV flow loop platform to simulate stable low blood volume (Figure 5 to Figure 9), how to reflect the advantages of the analysis method proposed in this article?
--Section4: Is the SACM method feasible in other closed-loop medical systems? Can it also achieve good analysis results? Does the author consider doing relevant trial work in the next research?
--Section4: Please consider whether the author can sort out and summarize the fourth part and add a summary part (Conclusions). So that readers can clarify the general conclusions obtained by the article research?
--Please refer to the basic requirements of the journal to modify the text and picture format.
Reviewer 2 Report
Answers for Authors
Dear Authors, thank you very much for submitting your paper "Supervisory Algorithm for Autonomous Hemodynamic Management Systems"
After reviewing heavily reviewing your paper i came to the following conclusions: The paper need to be revised.
The paper is overall of good merit but lacks some information, state of the art literature, structural refinements and some additional explanations. In the following i will display what needs to be added to the paper:
1. Overall Feedback:
- Revise all images and plots. Both the quality of the images (it is NOT a vectory graphic) and the size of the text in the axes-plots. The text in the images needs to be the same as in the normal text. In all the plots i can count the pixels, this is unnecessary
2. Abstract, Introduction and General Objective:
- The general objective is not clear. Please enter a list of clear contributions
3. State of the Art and Literature used:
- The state of the art is one of the biggest flaws in this paper because its very thin and its not covering the current papers in this field correctly. One of the foundations of a good scientific work is to include a holistic state of the art that covers the topic completely. Based on this an author should draw the conclusion for its own paper: What is new? What is the news value in my own paper? What are the contribution i will do in this paper?
I heavily encourage to search for more paper in the field.
5. Results:
- None of the images has a high quality and all of the images with the results are hard to read (i can count the pixesl). Please use a vector graphic for your results.
6. Discussion:
- good, nothing to add here.
7. Conclusions
- No conclusions or future work are presented here,
Author Response
Please see the attachment.

This manuscript is a resubmission of an earlier submission. The following is a list of the peer review reports and author responses from that submission.